# Direct Nanopore Sequencing of mRNA Reveals Landscape of Transcript Isoforms in Apicomplexan Parasites

V. Vern Lee,[a,b] Louise M. Judd,[c] Aaron R. Jex,[b,d] Kathryn E. Holt,[c,e] Christopher J. Tonkin,[b,f] Stuart A. Ralph[a]

[a]Department of Biochemistry and Pharmacology, Bio21 Molecular Science and Biotechnology Institute, The University of Melbourne, Melbourne, Victoria, Australia
[b]The Walter and Eliza Hall Institute of Medical Research, Parkville, Melbourne, Victoria, Australia
[c]Department of Infectious Diseases, Central Clinical School, Monash University, Melbourne, Victoria, Australia
[d]Faculty of Veterinary and Agricultural Sciences, The University of Melbourne, Parkville, Victoria, Australia
[e]London School of Hygiene and Tropical Medicine, London, United Kingdom
[f]Department of Medical Biology, The University of Melbourne, Melbourne, Victoria, Australia

**ABSTRACT** Alternative splicing is a widespread phenomenon in metazoans by which single genes are able to produce multiple isoforms of the gene product. However, this has been poorly characterized in apicomplexans, a major phylum of some of the most important global parasites. Efforts have been hampered by atypical transcriptomic features, such as the high AU content of *Plasmodium* RNA, but also the limitations of short-read sequencing in deciphering complex splicing events. In this study, we utilized the long read direct RNA sequencing platform developed by Oxford Nanopore Technologies to survey the alternative splicing landscape of *Toxoplasma gondii* and *Plasmodium falciparum*. We find that while native RNA sequencing has a reduced throughput, it allows us to obtain full-length or nearly full-length transcripts with comparable quantification to Illumina sequencing. By comparing these data with available gene models, we find widespread alternative splicing, particularly intron retention, in these parasites. Most of these transcripts contain premature stop codons, suggesting that in these parasites, alternative splicing represents a pathway to transcriptomic diversity, rather than expanding proteomic diversity. Moreover, alternative splicing rates are comparable between parasites, suggesting a shared splicing machinery, despite notable transcriptomic differences between the parasites. This study highlights a strategy in using long-read sequencing to understand splicing events at the whole-transcript level and has implications in the future interpretation of transcriptome sequencing studies.

**IMPORTANCE** We have used a novel nanopore sequencing technology to directly analyze parasite transcriptomes. The very long reads of this technology reveal the full-length genes of the parasites that cause malaria and toxoplasmosis. Gene transcripts must be processed in a process called splicing before they can be translated to protein. Our analysis reveals that these parasites very frequently only partially process their gene products, in a manner that departs dramatically from their human hosts.

**KEYWORDS** *Plasmodium*, RNA splicing, RNA-seq, *Toxoplasma*, nanopore, transcriptional regulation

Address correspondence to Stuart A. Ralph, saralph@unimelb.edu.au.

Direct RNA transcriptome analysis of eukaryotic parasites in a single nanopore flow cell.

Transcriptomic analyses have been central to insights into the biology and pathogenesis of eukaryotic pathogens. The best-characterized eukaryotic pathogen transcriptomes are those of the phylum *Apicomplexa*. This phylum includes some of the most important parasites impacting human and veterinary health, such as *Plasmodium* and *Toxoplasma*. *Plasmodium* is the causative agent of malaria, a devastating parasitic disease infecting over 200 million individuals and killing 400,000 each year (1).

*Toxoplasma* causes toxoplasmosis, a widespread zoonoses that primarily impacts immunocompromised, young, and pregnant individuals (2), and is thought to infect a third of the world's population (3). The pathogenesis of apicomplexan infections is intimately linked to the parasites' life cycles. The life cycle of most parasitic apicomplexans is complex, involving multiple differentiated forms and hosts, and this requires reprograming of the parasite transcriptome.

Early transcriptomic experiments sought to utilize techniques such as microarrays and Sanger sequencing of complementary DNA (cDNA) or expressed sequence tag (EST) libraries to understand changes in gene expression that define the pathogenesis of the parasites. These studies reveal that the timing of appearance and abundance of individual mRNAs follow developmentally distinct patterns (4), even for many predicted housekeeping genes. For example, the expression of the actin gene family in *Plasmodium falciparum* is developmentally tuned, with actin I primarily transcribed in asexual intraerythrocytic life stages, while actin II is primarily present in sexual-stage parasites (5, 6). Unusually, however, there is a poor correlation between protein and mRNA expression profiles for many genes in parasitic apicomplexans (7, 8). In one experiment, Foth et al. found widespread discrepancies between temporal expression patterns of proteins and transcripts in *P. falciparum* (9). Such discrepancies suggest that substantial posttranscriptional regulation occurs within these parasites. Indeed, with the advent of transcriptome/RNA sequencing (RNA-seq), more recent studies now show that multiple layers of gene expression regulation are required for parasite life progression, through transcriptional, posttranscriptional, and epigenetic control mechanisms (10–12).

RNA splicing provides one such source of co- and posttranscriptional regulation. In this process, introns are removed from the pre-mRNA and the exons retained to form one contiguous molecule that is then translated by the ribosome. However, for complex mRNAs, alternative splicing either of untranslated regions or the exonic chain can add additional complexity. Through this process, pre-mRNA species can be differentially spliced to create multiple distinct mature mRNAs from a single gene. This can alter regulation of the gene, e.g., by removing small-RNA binding sites (13), or diversify the proteome, as individual genes may encode multiple protein isoforms with altered structure or function (14). Indeed, proteomic analyses have revealed widespread protein isoforms arising from single genes, corresponding with various activity, stability, localization, and posttranslational modifications (15, 16). With advances in genome and transcript sequencing, it has become apparent over the last decade that alternative splicing of pre-mRNA occurs to a great extent. For example, more than 95% of human genes are alternatively spliced, and many transcript isoforms are specific to tissues or cellular states (17). Such observations suggest that RNA diversity is more complex than previously appreciated (18).

Although alternative splicing appears to play a major (though debated) role in posttranscriptional control in metazoans, the process is less understood in apicomplexans. Studies have identified apicomplexan genes with crucial alternative splicing outcomes (19). For example, alternative splicing is required for attaching a protein trafficking presequence onto two adjacent gene coding sequences (20), and normal multiorganellar targeting of the *P. falciparum* cysteinyl tRNA synthetase, which is essential for parasite survival (21). Nonetheless, there are few other studies of alternative splicing in this phylum. Understanding the diversity of parasites transcripts is crucial for drug and vaccine development because certain putative target genes may produce isoforms that escape the intervention. This has been postulated for the *P. falciparum* chloroquine resistance transporter gene (*Pf*CRT) in clinical isolates, though the role of the splice variants remains unclear (22). In other organisms, there is some evidence showing that essential genes are more likely to have alternatively spliced transcripts compared to nonessential genes (23, 24). This has not been explored in apicomplexans but highlights further considerations for investigating drug targets and interventions.

The lack of data for apicomplexan gene isoforms is a major obstacle to dissecting

the complexity of transcript outcomes. Traditionally, transcriptomic studies employing RNA-seq have relied on short-read technologies such as Illumina, 454, and Ion-Torrent (25). Despite the power of very high sequencing depth and low error rates, the short reads present a limitation in that simultaneously occurring alternative-splicing events within individual transcripts cannot be unambiguously detected or linked. Previously developed computational methods for full-length transcript assembly from short read sequencing data are often computationally intensive and can produce ambiguous or conflicting results between different algorithms (26). In addition, sequencing on cDNA strands amplified by PCR has a propensity to introduce biases in relative transcript abundances and rare isoform identification (27). Hence, it is difficult to draw functional relationships between simultaneous alternative splicing events and observable phenotypes. In apicomplexan parasites, simultaneously occurring alternative splicing events within a specific transcript isoform do occur (28). However, the studies that unearthed these transcript isoforms relied on cDNA probes and reverse transcription-PCR, and the wider extent of this phenomenon is unknown.

Recently developed third generation sequencing platforms, such as those developed by Oxford Nanopore Technologies (ONT) and Pacific Biosciences (PacBio), are capable of producing significantly longer reads at the single-molecule level. These technologies have been used in various applications such as resolving genomic and transcriptional landscapes (29, 30), single-cell transcriptome sequencing (31), and DNA or RNA methylation pattern profiling (32–34). PacBio has recently been used to generate an amplification-free transcriptome from *P. falciparum* cDNA, which has helped to elucidate transcriptional start sites and to improve annotation of the 5′ and 3′ untranslated regions (UTRs) (35). Unlike most other sequencing platforms, a notable characteristic of ONT sequencing is the ability to directly sequence native RNA (36). With this methodology, each read represents a complete molecular transcript, which could thus significantly resolve weaknesses of amplification-based RNA-seq. In particular, each spliced isoform need only be counted as individual reads, as opposed to complex assignment and assembly of multiple spliced reads. Furthermore, due to differences between DNA and RNA molecules, contaminating DNA sequences cannot be correctly base called after sequencing and so are easily discarded (37). Recently, several studies have successfully applied single molecule, long-read sequencing to identify a high number of novel transcript isoforms (29, 38, 39). However, these studies have also identified several caveats, including a reduced throughput and high error rates.

In this study, we evaluate the ability ONT direct RNA sequencing to characterize the alternative splicing landscape of two parasitic apicomplexans, *T. gondii* and *P. falciparum*. Our analyses show that alternative splicing, particularly intron retention, is extensive throughout the transcriptome, with most multi-exon genes having some degree of intron retention and some genes only rarely producing transcripts with all introns removed. The long reads produced from ONT sequencing showed that most of these alternative splicing events are likely nonproductive in protein-coding capacity but may provide an additional layer of gene expression regulation.

(This article was submitted to an online preprint archive [40].)

## RESULTS

**Direct RNA sequencing of *T. gondii* and *P. falciparum* allows the detection of full-length transcripts.** We generated ONT sequencing reads of poly(A)-selected RNA from asynchronous *T. gondii* (Prugniaud/Pru) tachyzoites and *P. falciparum* (3D7) mixed asexual intraerythrocytic-stage parasites. The choice to assay mixed-stage parasites, rather than specific, synchronized stages was dictated by the requirement for significantly more mRNA material than that required for short-read sequencing and the additional loss of yield when purifying full-length poly(A) mRNA. This is a limitation of this approach and means that we achieve a broader survey of transcript diversity present in asexual-stage replicative forms at the expense of stage-specific information. For *T. gondii*, we obtained a total of 310,813 reads corresponding to about 500 million bases (Mb). For *P. falciparum*, we obtained a total of 456,098 reads, corresponding to about

300 Mb of data. Although the *P. falciparum* sample yielded fewer sequenced bases, we estimate the theoretical gene coverage for both parasite samples to be similar at 25- to 26-fold due to differences in gene number and length. Using minimap2 (41), we successfully mapped 78.90% of the *T. gondii* reads and 44.48% of the *P. falciparum* reads to the parasite genomes. We analyzed the quality of the sequencing reads using FASTQC and found consistently high-quality scores over the length of reads, with no drop-off in quality even at reads of 10 kb (Fig. 1B). This is important because base quality scores generally correlate with read accuracy to the reference sequence. That is indeed the case for our data set (see Fig. S1A in the supplemental material). A few IGV snapshots of mapped reads are shown in Fig. S1B. We used AlignQC to estimate the base-call error rate of the transcript reads based on aligned segments and found, on average, an error rate of 19 to 20% for both parasites, although these numbers rely in part on the accuracy of the reference sequences. The read-length distributions of the mapped and unmapped data (Fig. 1A) show that the mapped reads are predominantly longer than the unmapped reads, with some read lengths exceeding 10 kb. As expected, there was a sharp increase in unmapped read counts at an ~1.35-kb length (Fig. 1A) corresponding to yeast enolase 2, a calibration standard added during the library preparation. We calculated the average mapped read lengths to be ~1.9 and ~1.3 kb for *T. gondii* and *P. falciparum*, respectively, values well within previous range estimates of predicted transcript lengths for both organisms (42).

To evaluate the capability of ONT sequencing to generate full-length transcript reads, we remapped the reads to the parasites' annotated transcriptome and calculated the fraction of full-length transcripts per gene. We define full-length transcripts as reads that cover more than 95% of the predicted canonical transcript based on the annotation file and only considered genes with at least three mapped reads. As illustrated in Fig. 1C, many transcripts were observed to have full-length reads, particularly for the *P. falciparum* data. In *T. gondii*, 1,117 genes have 75% or more of their corresponding reads that were considered full length. In *P. falciparum*, this number is 1835 genes. The difference can be attributed to the absence of 5' and 3' UTR annotations for *P. falciparum*, which results in the underestimation of predicted transcript lengths. In both cases, the fraction of mapped reads corresponding to full-length transcripts fell with increasing transcript length, independent of expression levels, which is consistent with read truncation disproportionately affecting the longest reads. To better understand the overall distribution of transcript coverage, we calculated the average coverage of the transcripts per gene. Again, there is a general decrease in transcript coverage with increasing transcript length (Fig. 1C). However, many of the genes retain a high level of transcript coverage, even when there is a low fraction of full-length reads. For example, in *T. gondii*, genes with predicted transcript lengths of 3 kb or longer only had an estimated 12% of their reads that were considered full length on average, even though the average read coverage for those genes is 50.64%. The overall average transcript coverage is calculated to be more than 60% in both parasites (*T. gondii*, 65.02%; *P. falciparum*, 80.51%). It should be noted that the relatively high number of genes with low expression (<10 reads) results in the overrepresentation of certain decimal values (observed as horizontal stripes on Fig. 1C). Only considering highly expressed genes improves coverage, but not significantly so (<5%). Together, the data indicate a generally high proportion of full-length or nearly full-length transcript reads.

**ONT sequencing is comparable with traditional RNA-seq for quantifying gene expression levels.** To investigate the utility of ONT data to measure transcript abundance, we computed read count correlations between our ONT data set and reanalyzed published Illumina-based RNA-seq data sets on comparable parasite samples. In theory, ONT RNA sequencing reads directly correspond to complete transcripts, so quantifying the expression of genes can be done by simple counting of the assigned reads. This is dissimilar to traditional short read RNA-Seq, which necessitates a further normalization step (e.g., reads or fragments per kilobase of transcript or transcripts per million) to account for the higher number of reads that would be generated from longer transcripts. For *T. gondii*, we used Illumina data sets from a closely related strain

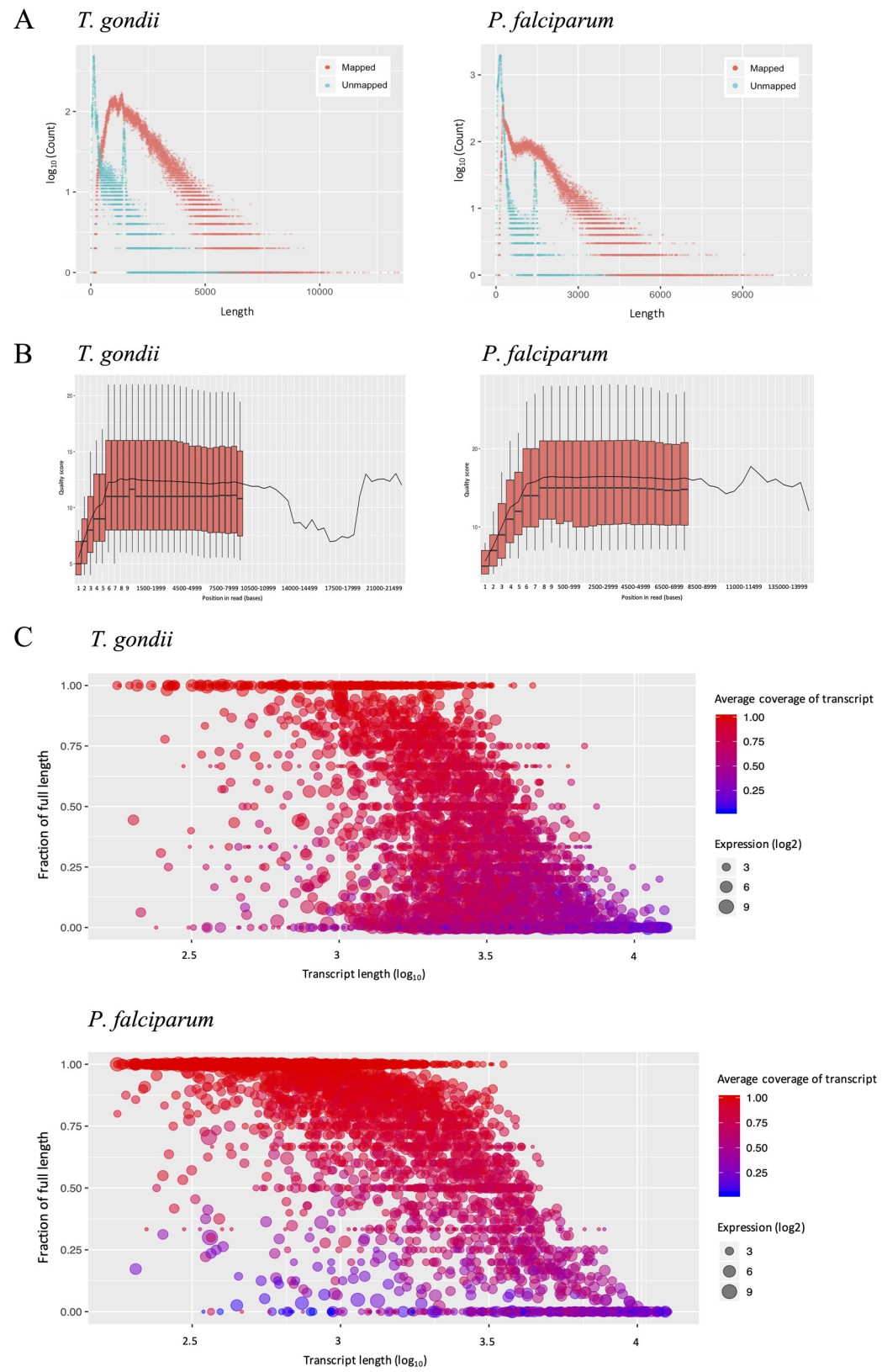

**FIG 1** Summary of the ONT direct RNA sequencing data from *T. gondii* and *P. falciparum*. (A) Scatterplot of read length distribution of mapped (red) and unmapped (blue) reads. (B) Boxplot of quality scores across all bases at each position of the mapped sequencing reads. Boxes signify interquartile ranges (25 to 75%), and whiskers represent the 10 and 90% points. The continuous line represents mean quality. (C) Bubble scatter heat plots of the fraction of full-length transcripts against transcript length. Size and color denote expression and average coverage, respectively.

(ME49) because it has the highest coverage replicated transcriptome data sets and is the closest well characterized strain (<0.01% genetic variation) to Prugniaud (43). Strikingly, we observe strong positive correlations between the ONT and Illumina data sets (Fig. 2), regardless of mapping to the transcriptome or genome (Spearman's rho = 0.81 for transcriptome and 0.87 for genome). For *P. falciparum*, we correlated the mixed-stage ONT data set with individual data sets from three main developmental stages (rings, trophozoites, and schizonts) and a final combined data set. In all cases, we found moderately positive correlations between the data sets. As expected, the correlation is higher in the later stages (see Fig. S2A; Spearman's rho = 0.47 for rings, 0.57 for trophozoites, and 0.63 for schizonts), and the highest with the combined data set (Fig. 2; Spearman's rho = 0.64 for transcriptome and 0.68 for genome), a reflection of mRNA abundance in these different stages.

For both parasites, a higher number of gene transcripts were detected in the Illumina data sets than in the ONT data (see Table S1). We detected at least 1 Illumina read for 8,379 genes in *T. gondii* and 5,639 genes in *P. falciparum*, as opposed to 6,778/4,656 genes for ONT reads. This is expected, given the greater sequencing depth that was obtained from the Illumina runs. The theoretical average fold coverages from our Illumina data are estimated to be 75 and 1,000 for *T. gondii* and *P. falciparum*, respectively (compared to 25- to 26-fold for the ONT data). Most of the genes that had detectable Illumina but not ONT reads were at the lower end of the expression range (see Fig. S2B). Notably however, abundant non-mRNA species, particular rRNA, were detected in the Illumina data sets. This is absent in the ONT data sets, possibly because of differences in poly(A) RNA purification methodologies and the fact that the sequencing adapters for ONT specifically ligate to poly(A) RNA. We further evaluated the ONT transcriptomes for genome coverage completeness, as shown in Fig. 2B. The read coverages from the ONT and Illumina reveal no significant bias toward a particular region of the nuclear genome. Of note, however, there appear to be high numbers of ONT reads mapping to the *P. falciparum* mitochondrion genome, but none to the apicoplast. This is expected and provides additional evidence that we are selectively capturing poly(A) RNA, since *P. falciparum* mitochondrion mRNA is typically polyadenylated (44), whereas apicoplast mRNA is not (45).

**Intron retained transcripts are prevalent and generally nonproductive.** A major goal of mature full-length transcript sequencing is the identification of splicing isoforms. Alternative splicing can be broadly classed into four types: intron retention, alternative 3′ splice site selection, alternative 5′ splice site selection, and exon skipping (46). Of these, intron retention is the least-studied form of alternative splicing despite the numerous studies implicating the significance of the event (47–49). This is in part due to the limitations of short-read sequencing but also the relatively long and low-complexity introns in metazoan genomes, which impose limitations on sequencing and assembly. For example, the intronic sequence in the human genome is several magnitudes longer than the length of the exonic sequence (50, 51). In contrast, the compact genomes of *Plasmodium* and *Toxoplasma* both have gene models that have similar or longer exon lengths than introns (52, 53), so reads that span multiple entire introns are quite achievable for these organisms.

To monitor levels of intron retention, we identified junctions and reads that overlapped annotated intronic regions of a gene based on the annotated coordinates using FeatureCounts (54), and we tallied the proportion of reads mapping to that intron to the total reads for the same gene. Proportion scores are represented using the metric percent intron retention (PIR). Based on this analysis, 17.65% of the mapped reads were considered to have intron overlaps for *T. gondii* versus 4.54% for *P. falciparum*. We further filtered out junctions without a minimum overlap of six bases to exclude artifacts generated by read errors and excluded genes that were not supported by a minimum coverage of three reads. The distribution of PIR scores per gene (Fig. 3A) reveals an overall skew toward low proportions of intron-overlapping reads. This is as expected given the propensity for a dominant canonical transcript (55). However, we

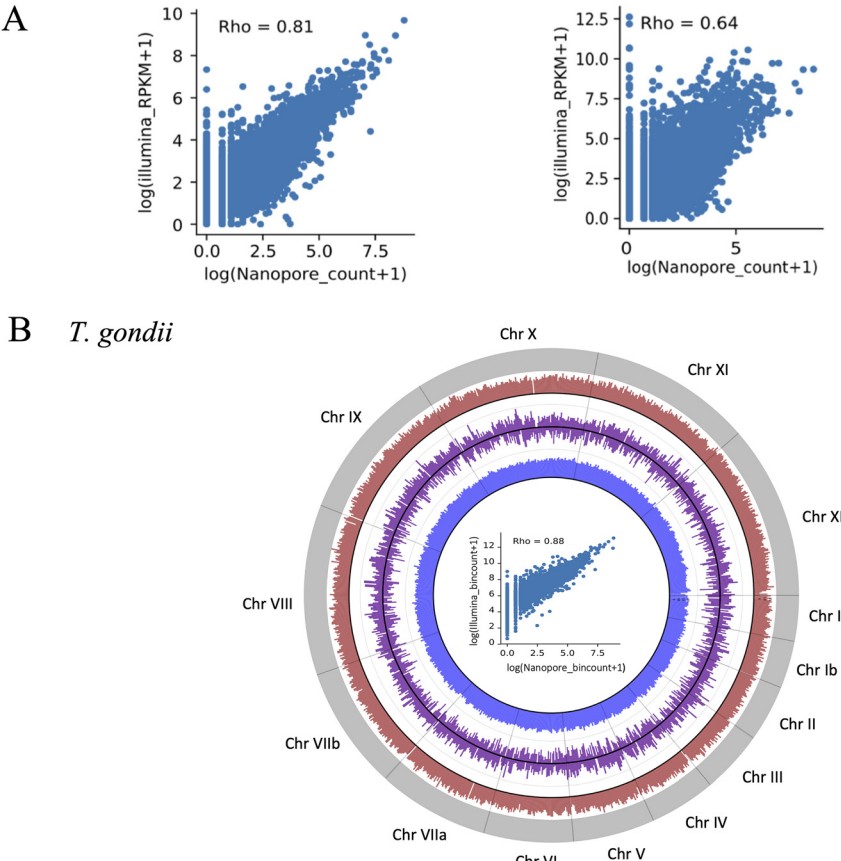

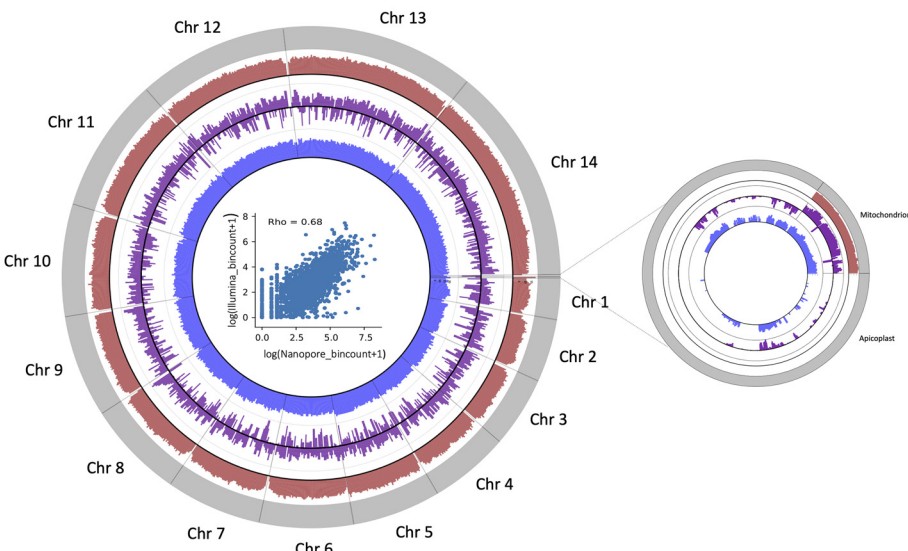

**FIG 2** Comparisons between ONT direct RNA sequencing and Illumina data sets. (A) Correlation between transcriptome mapped read counts for *T. gondii* (left) and *P. falciparum* (right), presented as a scatterplot. The Spearman correlation coefficient is shown. (B) Circos plots of genome-mapped reads. The outer band (gray) represents the reference genome/chromosomes. The red and blue bands represent the genome coverage of ONT direct RNA and Illumina reads, respectively. The purple band is the $\log_2$-fold change between the two data sets ($y$ axis limit = 20). The scatterplots within the Circos plots show the correlation between the genome-mapped read bin counts.

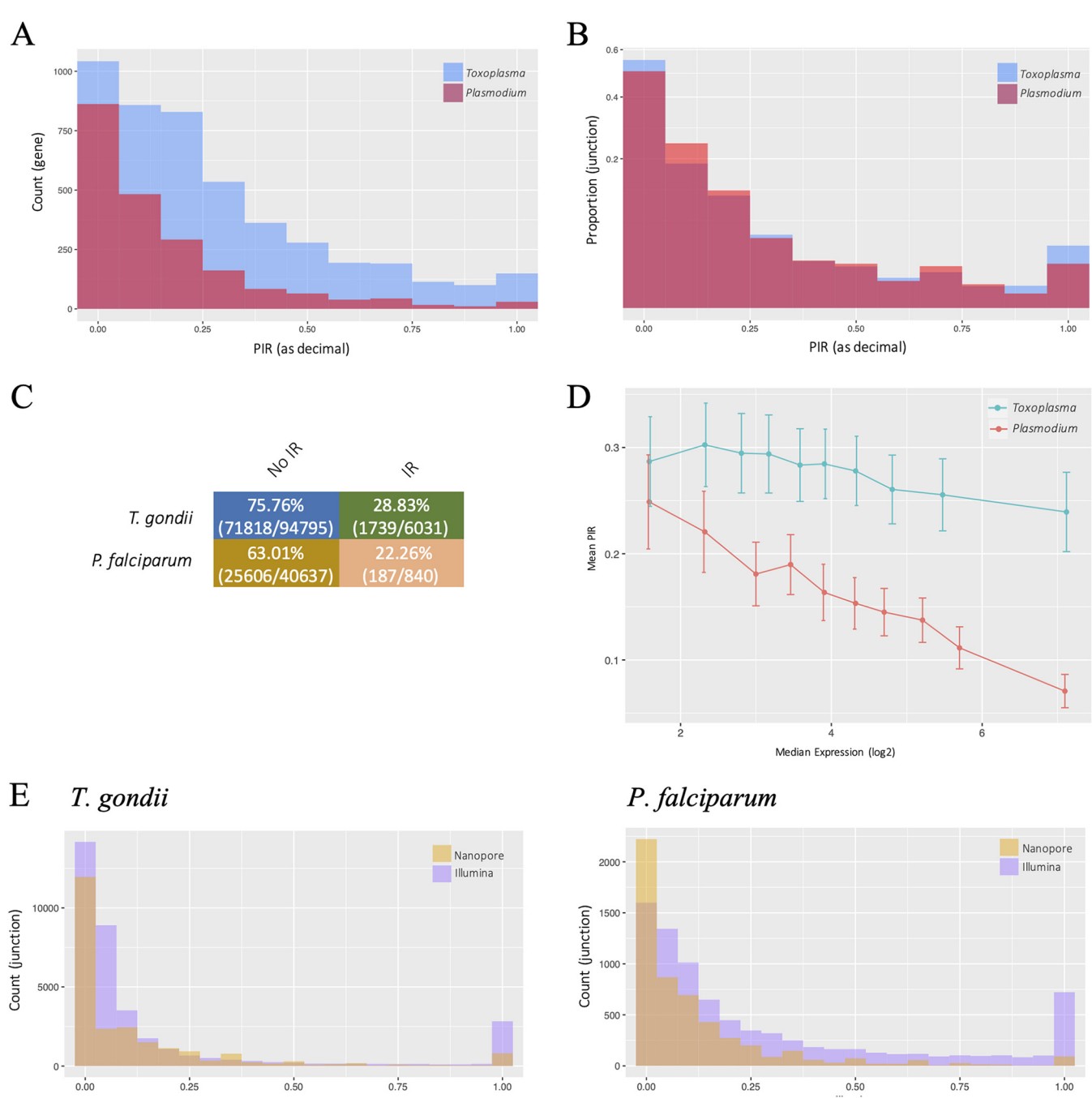

**FIG 3** Analysis of intron retention using ONT direct RNA sequencing reads. Levels of intron retention are represented as the percent intron retention (PIR). (A) Distribution of intron retention levels over each gene, represented as total counts. (B) Distribution of intron retention levels over each junction, represented as the proportion of each bin count over the sum all intronic junctions. (C) Table of transcript productivity based on intron retention events as analyzed using the FLAIR pipeline. Numbers in bracket are the transcript counts (D) Relationship between levels of intron retention and gene expression. Genes were classified into 10 bins of equal number based on expression. The median expression of the genes in each bin is used to represent expression for each bin. Error bars represent the 95% confidence intervals. (E) Comparisons of intron retention quantification between ONT and Illumina data sets.

identify a strikingly high number of genes that retain a high level of intronic regions. Using a threshold of 10% PIR, we identified a total of 3,229 of 4,653 (69.40%) expressed genes for *T. gondii* and 978 of 2,090 (46.79%) expressed genes for *P. falciparum* that have intronic reads within their transcripts (see Table S2A). Moreover, for approximately 29.82% (963/3,229) of the *Toxoplasma* genes and 19.63% (192/978) of the *Plasmodium* genes, 50% or more of the reads retain at least one intron.

When we only consider full-length or near-full-length reads (as previously defined),

mSystems®

the absolute quantification of intron retained genes is expectedly reduced, but the proportions of intron retained genes to expressed genes do not differ significantly (*T. gondii*, 1,325/1,887 [70.22%]; *P. falciparum*, 641/1,446 [44.33%]). Similarly, increasing the filtering threshold to only include genes with a minimum of 10 reads reduces absolute quantification but produces the same proportions (*T. gondii*, 2,085/2,988 [69.78%]; *P. falciparum*, 628/1,443 [43.52%]). This suggests that we are observing an inherent characteristic of the transcriptome rather than an artifact of the threshold that was used.

Unusually, there are a considerable number of genes where none of the transcripts appears to have all of their annotated introns removed (*T. gondii*, 133; *P. falciparum*, 29). We manually investigated these cases further and found major differences between the transcripts and gene model in most cases, suggesting that these highlight genes with an error in the existing gene annotation. A couple of examples are outlined in Fig. S1B. Most of these genes are annotated as hypothetical proteins, highlighting the potential of ONT sequencing to validate gene models.

The most extreme cases of conflict between the junctions we detect and canonical gene models often highlight potential annotation errors, but there are still a strikingly large number of genes where genuine introns are retained in a high (>50%) proportion of transcripts (*T. gondii*, 808 genes; *P. falciparum*, 162 genes). In addition, the differences between the two parasites are striking. In many organisms, the transcripts with the most introns are more likely to retain at least one or more introns (56). This is partially supported in our analysis, where we observed a higher level of overall intron retention for *T. gondii* (which has 4.5 introns per gene on average) than *P. falciparum* (which has only 1.5 introns per gene on average). A possible explanation for this relationship is thus that both organisms have a similar level of intron retention for any given junction, and the higher average intron number in *T. gondii* genes results in more overall intron retention per gene. To examine this, we calculated PIR scores at the individual junction level, rather than per-gene level, and we normalized the count of each PIR value to the proportion of total junctions within each organism. The analysis reveals that, after correction for the intron number, there is virtually no difference in the distribution of intron retention levels between parasites (Fig. 3B). In other words, individual *T. gondii* junctions are no more likely to experience intron retention than *P. falciparum* junctions. We further tested whether intron number was the major predictor of intron retention at the gene level by looking at the correlation between the number of introns per gene and levels of intron retention. Interestingly, we only obtained poor or moderate positive correlations in all data sets (see Fig. S3A; Spearman's rho = 0.28 for *T. gondii*, 0.48 for *P. falciparum*, and 0.39 for pooled samples). This correlation does not significantly improve even when we restrict our analysis to higher (≥10 reads) expressed genes. This suggests that whereas the intron number is associated with increased level of intron retention, it does not strongly influence whether a gene is alternatively spliced.

By taking advantage of the full-length reads made possible by ONT, we are able to predict the protein-coding productivity of the alternate transcripts. We performed productivity analysis on full-length intron-retained reads using the FLAIR (57) pipeline, which corrects and defines unproductive transcripts as transcripts with an annotated start codon and a termination codon that is 55 nucleotides or more upstream of the 3'-most splice junction. The rationale for this definition is based on previous evidence suggesting that only premature terminating transcripts following that 55-nucleotide rule mediate an effect on mRNA turnover (58). This is a conservative estimate of productivity since it does not consider intron retention within the 3'-most splice junction. The Flair method identified >70% of the intron-retained reads to be nonproductive for either parasite (Fig. 3C), suggesting that the high level of observed intron retention only rarely corresponds to alternative protein products, and that most intron-retaining transcripts may instead be targets for nonsense mediated decay. Intron retention is known to fine-tune protein expression through this pathway in mammalian systems

(59). A related prediction from other studies (47) is that the most highly expressed transcripts should have low levels of intron retention. In our analysis, we do observe a negative relationship between intron retention and gene expression levels, although it is less pronounced for *T. gondii* (Fig. 3D). There is a relatively high variance for this and more for genes with lower expression levels. This is likely due to the limitation in precision for the lower sequencing depths. For example, intron retention occurring in 10% of transcripts for a given gene will not be precisely measured for a gene for which only five reads are available. We circumvented this by classifying the transcripts into bins of equal read number based on expression and quantifying global intron retention levels within each bin. Again, we observe a negative relationship between intron retention and expression levels (see Fig. S3B). To investigate the functional significance of this, we further analyzed the genes for Gene Ontology (GO) enrichment. Here, we only considered genes with a minimum coverage of 10 reads to increase precision. The analyses reveal the consistent enrichment of genes with functions associated with the ribosome when there are lower levels of intron retention across both parasites (see Table S3). This association has been previously observed in other organisms (14), although its basis is unknown. We also tested whether intron retention correlated with essentiality based on previous functional genomic screens (60, 61) and found no significant relationships (see Fig. S3C).

To validate the identification of intron retention events, we looked at whether retained introns apparent in the ONT data were directly supported by Illumina RNA-seq data. We normalized read counts by junction length and only considered intronic data that spanned the full junction. Based on the analysis, 77.88% for *T. gondii* and 87.37% for *P. falciparum* of the intronic junction reads flagged from the ONT data sets were supported by Illumina reads. However, we also noted that some alternative splicing events, particularly the lower frequency ones, failed to be captured by ONT sequencing compared to the Illumina data set (Fig. 3E). When we applied the previously used threshold, we identified 1,272 intron retention events corresponding to 856 genes in *T. gondii*, and 2,008 intron retention events corresponding to 1,117 genes in *P. falciparum*, in the Illumina data set but not in the ONT data set. This is again likely due to the limitation in read depth in the ONT data set. Considering the theoretical fold coverage of the Illumina data set is around 4 times that of the ONT data set for *T. gondii* and 40 times for *P. falciparum*, we may expect to achieve increased levels of isoform quantification with increased number of sequencing runs/flow cells, although with diminishing returns.

Based on our sequencing of poly(A) tailed material combined with the sequencing chemistry of ONT direct RNA sequencing, which in theory only captures RNA with a poly(A) tail, and previous kinetic studies on RNA processing (62, 63), we do not expect the intron retained transcripts to simply be unprocessed transcripts. To confirm this, we looked for evidence that each transcript had at least been partially processed. On average, 92.76% of multi-intron genes identified as having intron retention within their transcripts had at least one junction that was canonically spliced in all the transcripts, demonstrating that our findings cannot be attributed to sequencing of pre-mRNA. To exclude the possibility that the efficacy of poly(A) tail selection of RNA in *P. falciparum* was reduced due to the high AU content throughout the RNA, we looked at the read coverage over each gene body, scaled to 100 nucleotides. This analysis reveals a strong 3′ end bias of reads (see Fig. S4), indicating a high efficacy of poly(A) tail RNA capture.

**Alternate junction splicing is often proximal and nonproductive.** Having previously identified intron-retained, read junctions using an annotated gene model approach, we used RSeQC (see Materials and Methods) to identify and quantify the other three classes of alternative splicing read junctions (exon skipping and 5′- and 3′-splice site changes) based on a similar methodology. The levels of alternative spliced junctions are calculated as the proportion of alternate junction over the total junction reads and are represented using the metric percent spliced (PS) value. Here, we filtered out junctions unsupported by a minimum coverage of three reads. Using the same

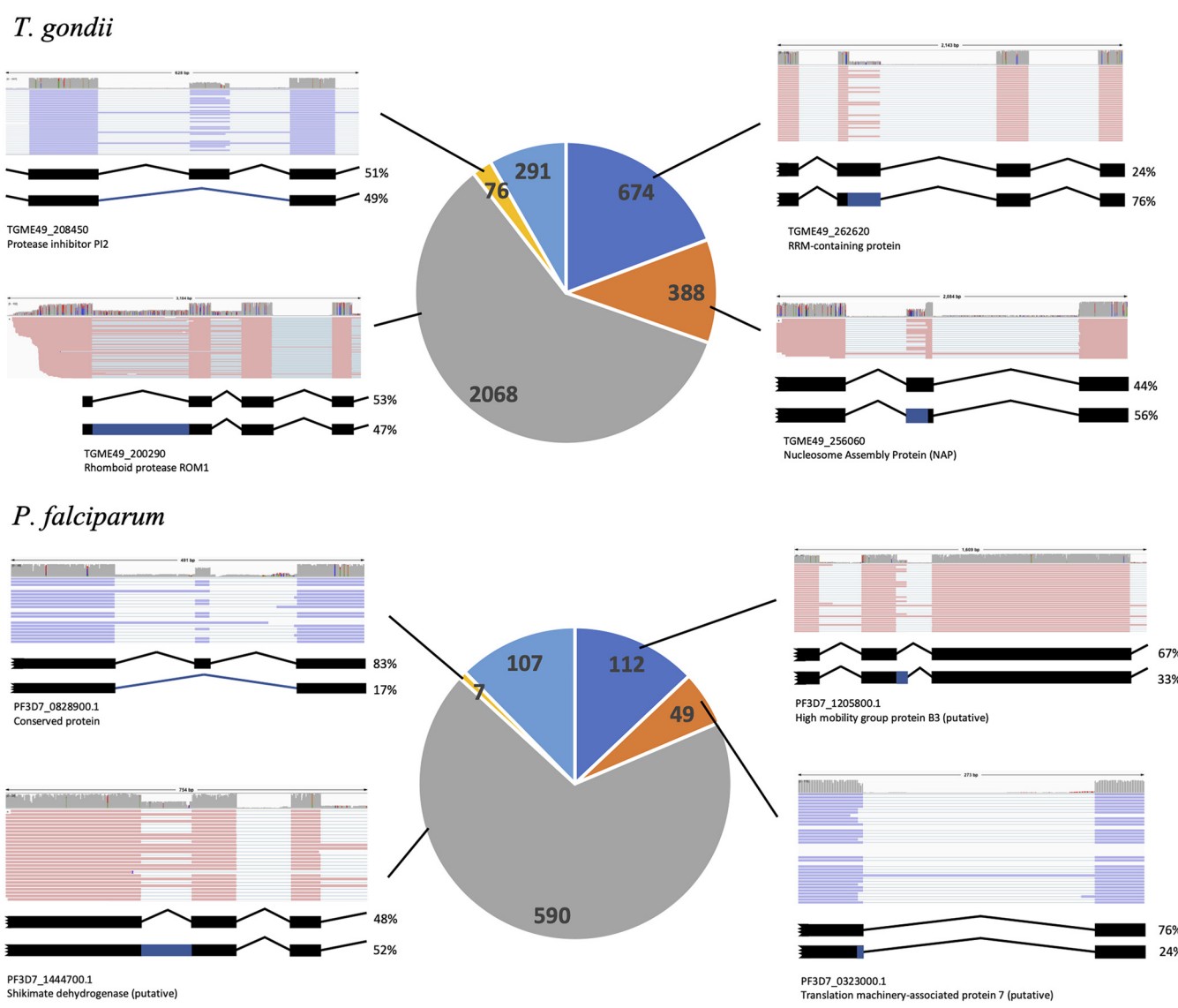

**FIG 4** Summary analysis of alternative splicing from the ONT direct RNA sequencing data sets. Pie charts show the number, proportion, and categorization of genes with alternatively spliced transcripts equaling or exceeding 10% of its total transcript. An example of each event is presented. Red and blue represents sense and antisense transcripts, respectively. Uncategorized genes represent genes where there are major mismatches between the RNAseq data and the annotated gene model.

threshold as before (≥10%), we identified a total of 1,138 genes for *T. gondii* and 168 genes for *P. falciparum*, where one or more of their junctions exhibited alternative 5′/3′-splice site selection or exon skipping (see Table S2B to D). Remarkably, these aggregate numbers are lower than those we calculated for intron retention alone. In total, we identified 5,205 splicing events for *T. gondii* and 1,112 for *P. falciparum*, and yet intron retention accounts for 60 to 68% of the alternatively spliced genes identified, alternate 5′ junction and 3′ junction splicing for 13 to 19% and 6 to 11%, respectively, and exon skipping for <3% (Fig. 4). The rest of the junctions flagged in the analysis defy easy categorization due to major mismatches between the RNA-seq data and the annotated gene model. Subsets of genes were also found to have multiple alternative splicing type events within their transcripts as observed in Fig. S5, although there does not appear to be a particular functional trend.

Having identified junctions subject to alternative splicing, we then quantified what

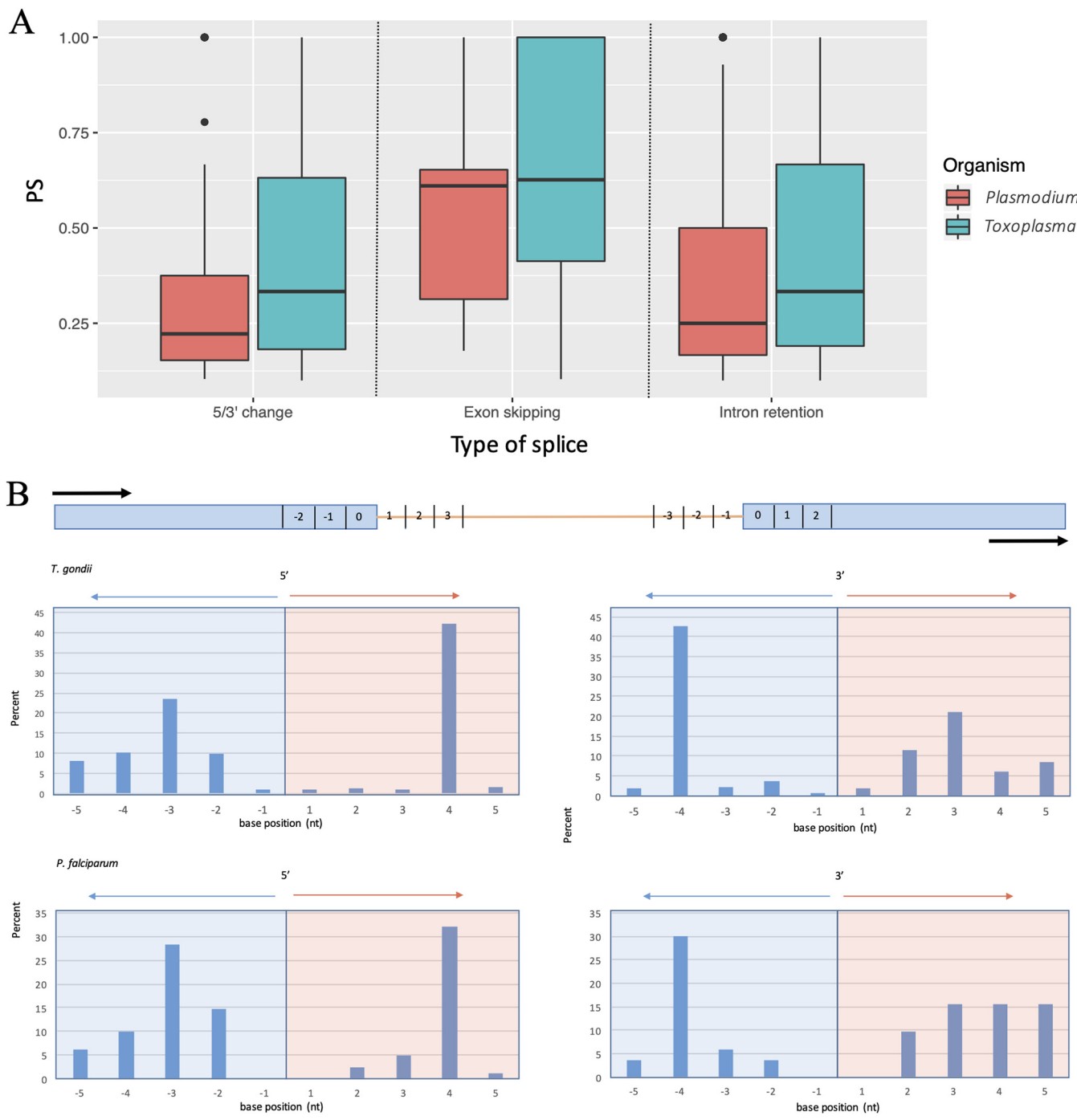

**FIG 5** (A) Levels of each major alternative splice type, represented as the percent splice (PS). (B) Distribution of alternate 5′/3′ splice positions within five bases to the dominant splice site.

proportion of the transcripts produced at those junctions represented the noncanonical isoform. For alternative splicing involving 5′ or 3′ changes and for intron retention, substantial proportions of isoforms were represented by the alternative transcript, but the median abundance remained below 50% (data shown in Fig. 5A). However, while exon skipping was a relatively rare event across the genome (Fig. 4), for genes where exon skipping did occur, it represented a higher proportion (approximately two-thirds) of transcript isoforms for those genes than observed for the other forms of alternative splicing (Fig. 5A). We had previously published a list of genes with alternative splicing excluding intron retention in *T. gondii* (RH) identified using Illumina sequencing and

our in-house-built computational tool JunctionJuror (64). A limitation of the tool over our current methodology is that the information about junction types is ultimately lost. Despite the differences in methodologies and parasite strain, we compared our list of genes with alternative splicing excluding intron with the previous list and found a moderately high degree of overlap (57.46%) (see Table S4). This intersect is double that is expected by chance alone and may represent genes with a steady-state alternative splicing of higher confidence. Similarly, we compared our data set with a previously published Illumina sequencing analysis which identified 310 alternative splicing events in blood-stage *P. falciparum* (65). There is an overlap of 49.75% alternative spliced genes between the data sets.

To further explore consequences of alternate 5′ or 3′ junction splicing events, we investigated the length distribution of the change in intron length. We found a surprisingly high proportion (~50%) of alternate 5′/3′ splicing to occur proximal (<6 bases) to the expected canonical site. We graphed the distribution of splicing change positions in Fig. 5B and found a substantial spike of splicing changes occurring at the position four bases inside the canonical intron boundary. We tested these junction reads for productivity for a subset of 50 of these "near-miss" alternate splice events and found all reads to prematurely terminate (unsurprisingly, given the necessary frameshift). This striking over-representation of isoforms departing from the canonical model by specifically four bases has been previously identified in metazoans (66). Very small movements in splice site usage have been described as junction wobbling, and this has been proposed as minor splicing noise, or alternatively, as an additional mechanism of regulation through the NMD pathway (66), although the reason for the specific peak of AS four bases away from the canonical junction is unknown.

## DISCUSSION

Despite the apparent significance of alternative splicing in metazoans, very little is known about the process in apicomplexans. A number of targeted experiments highlight single splicing events and their impact on parasite survival, but global splicing networks have been poorly described. Untargeted RNA-seq experiments have mainly focused on whole transcriptome assembly and/or gene expression. Those studies that do monitor alternative splicing reveal its occurrence in multiple genes and stages, but the extent of these events and their phenotypic significance remains unknown. The lack of a robust methodology in defining transcript isoforms from short-read data are a particular challenge in dissecting whole-transcriptome splicing. In this study, we investigated whether ONT direct RNA sequencing could be used to explore the splicing landscape of two apicomplexan parasites: *T. gondii* and *P. falciparum*.

To our knowledge, ONT direct RNA sequencing has not been previously described in apicomplexans and so our first objective was to evaluate the capability of ONT sequencing in generating sequencing reads from these parasites. We successfully obtained high-quality sequencing reads for both parasites that were comparable to that previously described in the literature for other organisms (29, 67). In particular, we obtained read lengths that exceeded 1 kb on average, many of them predicted to represent full-length or near-full-length transcripts. Interestingly, although we obtained a higher number of sequencing reads for *P. falciparum*, the mapping of the reads was suboptimal compared to that of *T. gondii*. Repetitive DNA sequence motifs are characteristic of many large eukaryotic genomes, and this has been known to complicate the mapping of reads that cannot be confidently assigned to these particular regions (68). In theory, long-read sequencing mitigates this problem because long enough reads should unambiguously match a unique site on the genome, irrespective of low complexity or repeat sequences. However, the genome of *P. falciparum* is particularly AT-rich (~82%), with numerous regions of extreme low complexity (69). Thus, we may expect some reads, particularly the shorter ones, to fail the mapping parameters. Indeed, as indicated above, reads that fail to map to the genome tended to be shorter than reads that do. This is further exacerbated by the high error rate produced from

ONT sequencing. Based on previous experiments, the per-base error rate of direct RNA sequencing using ONT is 10 to 20% (29, 57). In our data set, we estimated the error rate to be around 20%. This may have further contributed to the poorer mapping, though we do not expect the high error rate to significantly impact our study because the main analysis is focused on splice connectivity rather than on base sequences. As has happened for ONT DNA sequencing, we are likely to see significant improvements in read and mapping accuracy of RNA sequences as improvements are made to the flow cell and base caller. A study carried out by Runtuwe et al. provides an elegant example, wherein ONT DNA sequencing on targeted *P. falciparum* genes yielded a mapping percentage that improved from 57.86 to 92.46% with improved chemistry of the flow cell and upgrades to the base-calling algorithms (71).

The quantification of gene expression is one important goal of RNA-seq. Traditional, short-read sequencing requires the generation and amplification of cDNA, which can introduce artifacts and biases. Transcriptional amplification or repression is a commonly overlooked bias (72), where the levels of global mRNA, rather than specific mRNA, may vary between different samples. Thus, using a standard amount of total RNA, as is commonly done, can mask actual detection of specific mRNA levels, even after normalization (72). Direct RNA sequencing allows these caveats to be bypassed because a standard amount of isolated mRNA instead is used as the sequencing material. However, because there is no amplification step, direct RNA sequencing is limited by the amount of mRNA that can be practically obtained and used in the sequencing process. Without sufficient sequencing material, it can be difficult to achieve the high levels of sequencing depth that is needed to analyze gene expression (73). In line with the literature, our analysis shows that the current protocol for ONT direct RNA sequencing is comparable to Illumina for quantifying gene expression in the organisms we analyzed. It can be noted, however, that sequencing depth remains the main limitation of ONT sequencing in our study. The reduced throughput and sequencing depth from ONT sequencing compared to Illumina sequencing means that genes or transcript isoforms with low expression may not be captured.

Several previous analyses have reported differences in alternative splicing types and levels among different organisms (56, 74, 75). Notably, the increase in intron number and its retention correlates strongly with multicellular complexity (as defined by numbers of distinct cell types) (76, 77). In apicomplexans, the splicing machinery appears to be largely conserved, but features of gene structure such as intron number, length, and distribution can be highly variable (19). In our study, the difference in intron number between *T. gondii* and *P. falciparum* is a relevant example. Despite the differences, we found that alternative splicing for any given junction occurs at similar rates between the two parasites. This further supports the notion that the parasites share similar splicing processes. Intron number of genes is predicted to be positively associated with alternative splicing events (56, 78). This is not simply a stochastic effect but rather related to the general decrease in 5′ splice site strength with increasing intron number in many organisms (78). As described above, however, there is only a weak or moderately positive correlation between intron number and intron retention level of genes in the two parasites studied. Schmitz et al. (56) reported that other features such AT content and splice site entropy are important modulators of intron retention. This may also be the case in our study, given the observation that certain splice junctions are predisposed to retain their intron over others.

In addition to the difference in alternative splicing levels, there are differences in the composition of alternative splicing types between different organisms as reported by Kim et al. (75) and McGuire et al. (79). For example, exon skipping is the predominant form of alternative splicing in metazoans, and intron retention the rarest (75). Our analysis reveals that the opposite occurs in *T. gondii* and *P. falciparum*, with intron retention being the predominant event to occur and exon skipping the rarest. This composition of alternative splicing type is similar to that observed for plants and fungi (75, 79, 80), although the reason is unclear. More recent studies find that intron

retention has been previously under detected in metazoans due to methodology limitations or confounding variables, but the high levels of exon skipping has been mostly undisputed (81). Kim et al. (75) speculated that intron retention emerged as the earliest form of alternative splicing, before other mechanisms of complex splicing events evolved. There is some evidence for this, including the apparent shift toward increased exon skipping frequencies in early branching animals (81). This is associated with the preservation of coding frames, suggesting a role of exon skipping in expanding proteome diversity (81). In contrast to this, we found that the majority of the splicing events in *P. falciparum* and *T. gondii*, particularly intron retention, results in nonproductive transcripts. Our results thus indicate that alternative splicing rarely contributes to generating diversity of protein sequence in these parasites and may relate instead to transcriptomic complexity that impacts protein abundance. If that is true, a previous analysis that showed alternative splicing to be essential for *Plasmodium*-stage differentiation (12) may possibly be explained by a requirement for modulation of abundance for specific proteins rather than generation of protein sequences.

Consistent with our observation of alternative splicing playing a minor role in generating true proteome diversity in apicomplexans, many splicing events in other eukaryotes contribute little to the protein isoform repertoire. In particular, many transcripts contain premature termination codons (PTCs), at least in humans and yeast (82, 83). Often, PTC transcripts are the result of the retention of intronic sequences that contain PTCs (84), but translational frameshifts from active splicing events such as alternative splice site selections have been similarly implicated (83, 85). PTC transcripts are not normally translated but rather targeted for degradation through the nonsense-mediated decay (NMD) pathway (86, 87). This is vital because the transcripts encode altered or truncated proteins which may exhibit deleterious activity (88). Some studies postulate that the predicted alternative splice events are the result of either experimental or transcriptional noise (89) or that a substantial portion of such transcripts are contaminating pre-mRNA molecules and so do not represent true alternative splicing (59, 90). Nevertheless, many RNA-seq-based analyses operate on the assumption that PTC transcripts are biologically significant or relevant (91). Congruously, studies focusing on mature mRNA isoforms in other organisms suggest that nonproductive transcripts mediate an additional layer of posttranscriptional regulation, through downstream RNA processing changes such as mRNA turnover, export, and microRNA silencing (47, 92, 93). Alternative splicing in apicomplexans may also play a role in these processes. Strikingly, unique PTC transcript signatures are associated with distinct cell lineages (48, 94, 95) in multicellular eukaryotes, which may be analogous to the essential role of alternative splicing observed in stage differentiation observed in *Plasmodium* (12).

Nonproductive transcripts are typically degraded though the NMD pathway, and this has been shown to regulate gene expression at the posttranscriptional level (96). However, it is difficult to conclusively define the function of the nonproductive transcripts without experimentally testing the proteomic fates of these transcripts. In metazoans, nonproductive transcripts often highlight genes that were downregulated following a transition of cellular states (95). Our study is consistent with this, given the observation that the number of nonproductive transcripts generally decreased with increasing transcript number. This, in association with NMD, has been shown to be crucial to the maintenance and differentiation of many cell types (97, 98). In contrast, in organisms such *Paramecium tetraurelia*, nonproductive transcripts appear to be the result of splicing error rather than function (99). Regardless, because gene expression as measured by transcript levels do not necessarily translate to protein expression levels, our findings have potential implications for the interpretation of RNA-seq studies in these parasites. Several studies have already demonstrated the poor correlation between protein and mRNA expression profiles in apicomplexans (7, 9, 100). Our results highlight that for many genes raw quantifications of transcript abundance will correlate poorly with the number of copies of productive isoforms and provide one source of mismatch between transcriptional initiation and protein abundance.

Genome annotation is a crucial element of RNA-seq data analysis. For *T. gondii* and *P. falciparum*, the task is a widely accomplished manual effort from experts in the research community. Although genome annotation was not the main focus of the study, the ONT data sets are able to reveal the structure of full-length transcripts. This is crucial in validating gene models. Our data are viewable through the ToxoDB and PlasmoDB web resources (101), and raw data are available at the Sequence Read Archive (SRA; PRJNA606986), which may aid the research community to further curate and validate the current annotations.

**Conclusions.** In this study, we have performed the first direct transcriptomic analyses on *T. gondii* and *P. falciparum*. We show that ONT direct RNA sequencing enables the quantification of gene expression despite a reduced throughput. In combination with the increased requirement for starting material, this means that the cost and time per base pair sequenced remains higher than that of second-generation sequencing platforms. Nevertheless, because ONT direct RNA sequencing enables the detection of full-length transcripts without amplification, the tool remains promising for resolving the limitations of second-generation sequencing.

We demonstrated that alternative splicing is widespread in the two parasites, particularly intron retention. ONT direct RNA sequencing enabled us to determine the productivity of these transcripts without complex computational methodologies, and we show that most the transcripts are premature terminating. This has implications for the quantification of gene expression, since it is highly unlikely for the wealth of transcript diversity that we identified to directly translate to protein isoforms.

## MATERIALS AND METHODS

**Cell culture and RNA extraction.** *Toxoplasma gondii* tachyzoites (Pru Δ*ku80*) were cultured on human foreskin fibroblasts in Dulbecco modified Eagle medium supplemented with 1% (vol/vol) fetal calf serum and 1% (vol/vol) GlutaMAX. Freshly egressed tachyzoites were washed, filter purified (5 $\mu$m), and collected for RNA extraction. *P. falciparum* (3D7) were cultured in complete media consisting of human erythrocytes (O+, 2% hematocrit), RPMI-HEPES, 5% (wt/vol) Albumax, and 3.6% (wt/vol) sodium bicarbonate. We collected mixed-stage parasite, purified from host red blood cells via lysis with 0.05% (wt/vol) saponin, for RNA extraction. To obtain the 500 ng of mRNA recommended for the library preparation, we used TRI Reagent (Sigma) for extraction of total RNA followed by the Dynabeads mRNA purification kit for polyadenylated [poly(A)] mRNA (Thermo Scientific) purification according to the manufacturer's protocol. Purity and quantification of mRNA were determined via NanoDrop (Thermo Scientific) and a Qubit RNA HS assay kit (Thermo Scientific).

**Library preparation and Nanopore sequencing.** Libraries for the direct RNA sequencing were generated using the recommended protocol for the SQK-RNA001 kit (Oxford Nanopore Technologies). We loaded and sequenced the libraries on MinION R9.4 flow cells (Oxford Nanopore Technologies) for 48 h. Base calling was performed concurrent with sequencing using Albacore (v 2.0), which was integrated within the MinION software (MinKNOW, v1.10.23). Only "pass" reads as designated by the tool were used for subsequent analyses.

**Mapping and qualitative analysis.** ONT sequencing data were first checked for quality with FastQC (v.0.11.7) (102). We then utilized Minimap2 (v2.1) (41) to map raw reads to the parasite genome and transcriptome from ToxoDB and PlasmoDB (r. 39), using the recommended preset commands except that intron length thresholds were set at 5,000 and 1,500 bases for *T. gondii* and *P. falciparum*, respectively. Previously published Illumina data sets (SRR350746, ERR174301, ERR185969, ERR185970, and ERR185971) (103, 104) were mapped using HISAT2 (105) using the preset commands. We checked for mapping quality with SAMtools (v.1.7) (106), Picard (v.2.18.2) (107), and AlignQC (v.1.2) (108). Further postprocessing of data was performed using the command-line interface, and graphical elements were drawn using ggplot (109) on RStudio. We verified and illustrated subsets of mapped reads via IGV (110).

We correlated the ONT sequencing data with the Illumina data sets as previously described (36) using the wub package (v.0.2) (111). The genome coverage of sequencing data sets were generated using bedtools genome coverage (v2.27) (112) and visualized via J-Circos (v1) (113). Log$_2$-fold ratios were calculated using deepTools bamCompare (v.2.5.1) (114). Gene body coverage was analyzed with the geneBody_coverage.py script of RSeQC (v.2.6.4) (115) using the default settings. For additional details and command lines used in these analyses, see Text S1 in the supplemental material.

**Alternative splicing analysis.** We applied two approaches to analyzing alternative splicing. We first identified intron retained junctions and transcripts using featureCounts (v.1.6.2) (54) on the genome mapped reads. featureCounts matches features specified in an annotation file (gff) to mapped reads. The annotation files used in the analyses were obtained from ToxoDB and PlasmoDB (r.39), and preprocessed via ToolShed (v.1.0) (116) to specifically extract intron coordinates and gene IDs. We set a minimum threshold requiring mapping to at least six bases of the intron feature and a minimum threshold of three reads mapping to the junction/transcript to be considered for further analysis. PIR scores were calculated as the proportion of alternative splicing events to the sum of reads for each junction/gene.

Productivity of full-length transcripts was analyzed using the Flair pipeline (57) using default parameters.

For the second approach, we used the junction_annotation.py script of RSeQC (v.2.6.4) (115) to identify novel or partial-novel junctions from the genome mapped reads based on the unmodified annotation file. Again, we filtered out junctions that had fewer than three supporting reads. The junctions were summarized into a table based on coordinate matching to the 5′ and/or 3′ of the expected canonical junction. We identified alternative 5′/3′ splicing and exon skipping based on the coordinates and strandedness of junctions identified by RSeQC that were either consistent with or conflicted with the annotated junctions. We manually validated the data, matched junctions to available gene IDs, and again calculated the proportion of alternative splicing events to the sum of reads for each junction. Using the final data set, we re-curated the list of intron-retained junctions to exclude for alternate 5′/3′ splice changes. Proportional Venn diagrams were drawn using BioVenn (117).

Gene set enrichment analyses were carried out by ranking the genes based on their alternative splicing levels and using the first and third quartiles of the ranked list as input for GO enrichment analysis via ToxoDB/PlasmoDB based on curated and computed assigned associations. We required the adjusted $P$ value to be <0.05 and the false discovery rate q-value to be <0.25. This approach was validated using GSEA via WebGestalt (118).

**Data availability.** Our data are viewable through the ToxoDB and PlasmoDB web resources (101), and raw data are available from the SRA (PRJNA606986).

## SUPPLEMENTAL MATERIAL

Supplemental material is available online only.

**TEXT S1**, DOCX file, 0.02 MB.
**FIG S1**, TIF file, 2.5 MB.
**FIG S2**, TIF file, 2.3 MB.
**FIG S3**, TIF file, 2.6 MB.
**FIG S4**, TIF file, 1 MB.
**FIG S5**, TIF file, 0.9 MB.
**TABLE S1**, XLSX file, 0.6 MB.
**TABLE S2**, XLSX file, 0.7 MB.
**TABLE S3**, XLSX file, 0.1 MB.
**TABLE S4**, XLSX file, 0.1 MB.

## ACKNOWLEDGMENTS

This research was supported by a grant from the Australian Research Council (DP160100389) and the Australian National Health and Medical Research Council (Project Grant 1165354). The funders had no role in study design, data collection and interpretation, or the decision to submit the work for publication.

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
