## [Reviewer comments · mSystems]

Direct nanopore sequencing of mRNA reveals landscape of transcript isoforms in apicomplexan parasites

V Lee, Louise Judd, Aaron Jex, Kathryn Holt, Chris Tonkin, and Stuart Ralph

Corresponding Author(s): Stuart Ralph, University of Melbourne

Review Timeline:

Submission Date:	October 19, 2020
Editorial Decision:	November 23, 2020
Revision Received:	February 1, 2021
Accepted:	February 6, 2021

Editor: Paola de Sessions

Reviewer(s): Disclosure of reviewer identity is with reference to reviewer comments included in decision letter(s). The following individuals involved in review of your submission have agreed to reveal their identity: Scott Lindner (Reviewer #3)

Transaction Report:

DOI: <https://doi.org/10.1128/mSystems.01081-20>

November 23, 2020

Dr. Stuart A Ralph
University of Melbourne
Department of Biochemistry and Molecular Biology
Bio21 Molecular Science and Biotechnology Institute
Parkville 3010
Australia

Re: mSystems01081-20 (Direct nanopore sequencing of mRNA reveals landscape of transcript isoforms in apicomplexan parasites)

Dear Dr. Stuart A Ralph:

Lee et al. demonstrate for the first time Nanopore sequencing of two apicomplexans, and show that alternative splicing, in particular intron retention, is widespread for both of them. You also manage to provide evidence that most of these alternative splicing events are not protein coding. Given that these tropical neglected parasites have not been sequenced on the Nanopore platform before, we feel that this publication would interest the mSystems readership greatly. The authors have been able to reach interesting conclusions despite the low coverage of your sequencing runs.

Below you will find the comments of the reviewers.

To submit your modified manuscript, log onto the eJP submission site at <https://msystems.msubmit.net/cgi-bin/main.plex>. If you cannot remember your password, click the "Can't remember your password?" link and follow the instructions on the screen. Go to Author Tasks and click the appropriate manuscript title to begin the resubmission process. The information that you entered when you first submitted the paper will be displayed. Please update the information as necessary. Provide (1) point-by-point responses to the issues raised by the reviewers as file type "Response to Reviewers," not in your cover letter, and (2) a PDF file that indicates the changes from the original submission (by highlighting or underlining the changes) as file type "Marked Up Manuscript - For Review Only."

Due to the SARS-CoV-2 pandemic, our typical 60 day deadline for revisions will not be applied. I hope that you will be able to submit a revised manuscript soon, but want to reassure you that the journal will be flexible in terms of timing, particularly if experimental revisions are needed. When you are ready to resubmit, please know that our staff and Editors are working remotely and handling submissions without delay. If you do not wish to modify the manuscript and prefer to submit it to another journal, please notify me of your decision immediately so that the manuscript may be formally withdrawn from consideration by mSystems.

Corresponding authors may join or renew ASM membership to obtain discounts on publication fees.

Need to upgrade your membership level? Please contact Customer Service at Service@asmusa.org.

Sincerely,

Paola de Sessions

Editor, mSystems

Journals Department
Reviewer comments:

Reviewer #1 (Comments for the Author):

Major comments:

- 1.The authors should characterize their sequencing data so that readers can understand the data generated better. One concern is that the sequence coverage is low at only one flow cell used per parasite transcriptome. To let readers understand the pros and cons of sequencing more, are the authors able to reveal more about the isoforms or genes not detected from their nanopore sequencing runs? It would be good if they could extrapolate the results of their nanopore sequencing runs to they could let readers know how many flow cells to run to cover a given percentage of the parasite transcriptome.
- 2.Can the authors explain why for Figure 1C and 1D, there seems to be transcripts that have the same fraction of full length reads, across a range of different transcript lengths? This is seen as a horizontal collection of data points that have the same y axis value. For example in Figure 1D, we see this occurring when the fraction of full length reads is 0.25, 0.30 (approximately), 0.50 and 0.70 (approximately).
- 3.For *P. falciparum*, what is the point of comparing the mixed stage nanopore dataset with the three main developmental stages? We already know that the *P. falciparum* data is a mix of the three and so the correlation can be expected to increase with mRNA abundance. I would have thought a better comparison would be to compare nanopore versus illumina datasets for the three developmental stages in isolation, and not to mix them in the first place. After all the point of this section is to compare the two sequencing modalities' similarity in measuring gene expression levels. Mixing the three developmental stages does increase transcript diversity but we lose the ability to link isoform differences to the developmental stage, which is quite a pity.
- 4.For the analysis of intron retention, do the results of the analysis change when we only consider full-length reads? This section also seeks to associate the effect of intron number with intron retention, and concludes that intron number "is not the main determinant of whether a gene retained at least one intron". I do not think that we can conclude that intron number is the main determinant of whether a gene retained at least one intron, even with a strong correlation. Suggest

rephrasing this. Another point is that the filtering of junctions excluded genes that were not supported by a minimum coverage of three reads. Would the conclusions hold if the filtering threshold is made more stringent?

Minor comments:

1. What would the justification be for the use of the ME49 strain Illumina dataset as compared to sequencing the Pru strain on the Illumina platform, besides a relative lack of comprehensive datasets? This is for a fair comparison as opposed to a ballpark, as the reasoning was that the ME49 strain is closely related to the Pru strain, but the degree of similarity between the Pru and ME49 transcriptomes has not been pointed out.
2. The point about rRNA reads present in Illumina versus nanopore datasets- would this be due mainly to the fact that the RNA was directly sequenced and therefore sequences without polyA tails were not sequenced, and not what the authors point out to be differences in polyA purification or PCR amplification bias?

Reviewer #2 (Comments for the Author):

The paper by Lee et al. entitled "Direct nanopore sequencing of mRNA reveals landscape of transcript isoforms in apicomplexan parasites" describes the results obtained by using Oxford NanoPore Seq of two Apicomplexan parasites: *Toxoplasma gondii* and *Plasmodium falciparum*. The main finding of this ms is that intron retention is the most common alternative splicing event in these parasites.

The paper is clearly written (in most parts), nicely structured and the data is well presented. Introducing Nanopore sequencing, which enables analysis of long transcript reads has obvious potential for detection of alternative splicing events as the outcome is entire or almost complete transcripts.

It also had the potential to provide a valuable comparison between Illumina and nanopore sequencing for analyzing alternative splicing events in *T. gondii* and *P. falciparum*. However, the experiments performed seem preliminary. They present the results of only one run of unsynchronized parasite culture. The quality of the data obtained appears to be low, less than 50% of the Plasmodium transcripts aligned properly to the reference genome. The threshold used is 3 transcripts per gene which is very low. As such, it is hard to perform proper comparison to Illumina technology (I also could not find the exact reference for the Illumina data set used).

From a biological aspect the paper does not add much to what was already known (intron retention appeared also to be the most abundant alternative splicing event in previous publications). Few of the previous publications (some already 10 years old) that used Illumina technology and first describe the 977 alternative splicing events in blood stage parasites (more than what was found in the current papers) are not mentioned.

Reviewer #3 (Comments for the Author):

Lee and colleagues here present a first comparison of Illumina short-read sequencing and Oxford Nanopore Technologies (ONT) direct RNA long-read sequencing with two apicomplexan parasites: *Toxoplasma gondii* (Tg) and *Plasmodium falciparum* (Pf). Using readily produced Tg tachyzoites from the Pru strain, and Pf mixed blood stages from the common lab strain 3D7, the authors put these samples through the manufacturer's recommended workflow and compare these data to published Illumina datasets. Using these comparisons, they focus on differential splicing events that

occur in these related apicomplexan parasites, which is a strength of the ONT sequencing platform due to the long reads it provides. As has been seen for other eukaryotes, these long reads often provide full-length (or nearly full length) transcripts and provide a wealth of direct information about splice isoforms. In contrast, Illumina short-read sequencing must rely upon predictive algorithms to anticipate splice isoforms.

One of the major findings presented in this manuscript is the greater abundance of intron retention events as compared to other eukaryotes. What is not clear here is whether all of these mRNAs are those that have completed nuclear processing and have passed quality control to be exported to the cytosol, or if these represent mRNAs that are still resident in the nucleus and are currently undergoing maturation. While poly(A) selection will promote capture of mature mRNAs in most species (as polyadenylation of transcripts typically follows after co-transcriptional capping and splicing), the higher A/U content of Plasmodium mRNAs reduces the efficacy of this approach. It would be important to describe how this possibility is excluded (experimentally or otherwise) when other eukaryotes have been studied.

Generally, the sequencing, mapping, and comparison efforts are of very high quality. Most figures are clear and present the data in a concise manner. Comments are provided below that will further clarify the manuscript so that readers more readily follow the data and interpretations, and so that the experiments and analysis can be accurately repeated by others. This first foray into ONT direct RNA sequencing of apicomplexans will be a solid study for mSystems.

Major Points:

- It appears that all of this data is generated from sequencing of a single biological replicate. Most transcriptomic studies require multiple biological replicates for rigor.
- As noted above, intron retention could reflect the sequencing of pre-mRNAs that are still undergoing maturation in the nucleus. Additional discussion about how this could be excluded (if appropriate), the extent that this is observed in transcriptomes of other eukaryotes, or other qualifiers would be helpful. This is especially important due to the claim that this type of splice isoform is more abundant in apicomplexans as compared to other eukaryotes.
- Expansion of the bioinformatic processing of the ONT sequencing data is needed so that this work can be fully replicated. For instance, no information is provided as to what Q score threshold is used to designate "pass reads," which has implications for other analyses in this work. It wasn't explicitly mentioned if adapters were trimmed (not doing so would make the data quality lower). Other tools are mentioned, but their use/parameters are not described in detail. An expansion of the explanation of the statistical analyses done using command line/RStudio and where their implementation as represented in the results section is warranted.
- Inclusion of a description of all methods used in the Results is needed.

Minor Points:

- Line 22: Should be AU content, as this is RNA.
- Line 38: The claim that this is a complete analysis of parasite transcriptomics in the "Importance" section is of course a bit too large and broad. This is an important first step, but many other aspects of parasite biology remain to be explored. Please temper this claim.
- Additional/different citations are needed in a few critical locations. This includes:
 - 1) providing the citation for the published Illumina datasets (e.g. in the methods section) is helpful to include in addition to providing the SRA accession numbers.
 - 2) Line 51: Citing the most recent World Malaria Report would be better than a review from a few years back.
 - 3) Line 66: only a citation for Plasmodium is provided here, but a statement is made to encompass apicomplexans.
 - 4) Line 125: Providing a citation on detecting modified bases by ONT direct RNA sequencing in the

Introduction is warranted as well (e.g. Liu et al. Nature Communications 2019 may be suitable here).
5) Line 128: Please confirm that Ref 33 is not published (this is a biorxiv preprint from 2019).
-Line 171 and/or Line 422-423: It would be useful to hear more speculation as to why the error rate was so high in these datasets. Could it be due to not trimming adapter sequences? Something else?

Figures and Tables:

- Figure 1B: Designation of error bars is not provided (confidence intervals?). Also, the red box plots are not provided for reads >8000, but Quality scores are.
- Figure 2A: These comparisons are not with matched samples, so it is difficult to glean much from these scatter plots.
- Figure 2B: No y-axis (radial) values are provided, so the circos plots are not informative enough. An inset of the mitochondrial RNA (currently not shown) and the apicoplast RNA (not shown in a useful way) would be helpful.
- Figure 3B: There is an additional color in the bar chart (light red) that is not designated.
- Figure 3C: Are tables allowed in Figures for mSystems?
- Figure 3D: As noted above, the negative correlation is clear for Plasmodium, but not for Toxoplasma. Please solidify or temper this claim.
- Figure 4: It would be helpful to note the types of events that are being binned as "uncharacterized" here.
- Supp Figure 1: The reported Quality score range here is discordant with the 20% error rate reported.
- Supp Figure 2: These comparisons are not very useful, as they are comparing mixed stage parasites with one sequencing method with synchronized parasites with the other. Consider omitting this.
- Supp Figure 3: It is likely that this is just a bad annotation for these two genes. Are there other examples of this?
- Supp Figure 4: The blue color on the right is not the same as the legend.
- Supp Figure 5: There are no confidence intervals provided here, whereas they are provided in a similar figure in the main text.
- Supp Table 3: The Plasmodium tab is blank.
- Supp Table 6: This is not referenced in the manuscript, and there is no legend for it either. It appears to be just a gene list.

Typos/Grammar:

Line 22: Italicize Plasmodium

Line 29: "particularly intron retention" instead of particular intron retention

Line 90: "is" instead of "in"

We sincerely thank editor and the referees for their careful review and constructive comments. We have made extensive changes to the text and figures and have generated additional figure elements and supplementary figures to address the comments of the referees, and believe that this has considerably improved the manuscript. We have responded point-by-point to the referees' comments below. We have included a version of the manuscript with all changes marked as well as a clean version. We hope that the manuscript is now ready for publication.

Reviewer #1

Major comments:

1. The authors should characterize their sequencing data so that readers can understand the data generated better. One concern is that the sequence coverage is low at only one flow cell used per parasite transcriptome. To let readers understand the pros and cons of sequencing more, are the authors able to reveal more about the isoforms or genes not detected from their nanopore sequencing runs? It would be good if they could extrapolate the results of their nanopore sequencing runs to they could let readers know how many flow cells to run to cover a given percentage of the parasite transcriptome.

- The referee makes a good point that the lower sequence output of nanopore compared to other technologies (particularly illumina) is likely to result in lower fold coverage for similar amounts of parasite starting material – however, part of the higher sequence depth achieved in other technologies relies on amplification of the starting material, and not necessarily improved characterization of the real transcriptome landscape. A higher fold coverage with short reads may not result in the same detection of transcript isoforms as a lower fold coverage with the long reads possible with nanopore sequencing. However, using very high coverage illumina sequencing will indeed detect more transcripts than using a single flow cell for nanopore sequencing, particularly for genes with very low expression levels and we have now included a more direct comparison of these data at lines 242-243, 247-248, and 376-383, as well as an additional supplementary figure (S2B) describing the transcripts that are detected with one sequencing technology but not the other.

2. Can the authors explain why for Figure 1C and 1D, there seems to be transcripts that have the same fraction of full length reads, across a range of different transcript lengths? This is seen as a horizontal collection of data points that have the same y axis value. For example in Figure 1D, we see this occurring when the fraction of full length reads is 0.25, 0.30 (approximately), 0.50 and 0.70 (approximately).

- This is due to a bias in the number of genes with few transcripts (less than 10), which results in the over-representation of certain decimal values. We have now made this clear in lines 210-213. Excluding transcripts with low abundance does not actually alter the results (for average coverage) which is why we have included them.

3. For *P. falciparum*, what is the point of comparing the mixed stage nanopore dataset with the three main developmental stages? We already know that the *P. falciparum* data is a mix of the three and so the correlation can be expected to increase with mRNA abundance. I would have thought a better comparison would be to compare nanopore versus illumina datasets for the three developmental stages in isolation, and not to mix them in the first place. After all the point of this section is to compare the two sequencing modalities' similarity in measuring gene expression levels. Mixing the three developmental stages does increase transcript diversity but we lose the ability to link isoform differences to the developmental stage, which is quite a pity.

- Taking into consideration cost-benefit, we are hampered by the significantly higher amounts of RNA material needed for ONT sequencing. For comparison, a typical Illumina sequencing run requires less than 500 ng total RNA. The Nanopore run required 500 ng of just poly A RNA, which makes up 2-3% of total RNA. This substantially limited our ability to purify early (ring) stage *Plasmodium* forms, so we elected to harvest mixed stages to maximise the diversity of transcripts surveyed. For this reason, we would not particularly advocate the use of nanopore for fine scale analysis of stage-specific changes in gene expression. We have now edited lines 162-166 to make this clearer.

4. For the analysis of intron retention, do the results of the analysis change when we only consider full-length reads? This section also seeks to associate the effect of intron number with intron retention, and concludes that intron number "is not the main determinant of whether a gene retained at least one intron". I do not think that we can conclude that intron number is the main determinant of whether a gene retained at least one intron, even with a strong correlation. Suggest rephrasing this. Another point is that the filtering of junctions excluded genes that were not supported by a minimum coverage of three reads. Would the conclusions hold if the filtering threshold is made more stringent?

- This is an excellent point – while we were already looking at nearly full length reads (as defined by the FLAIR analysis), we tested the sensitivity of this analysis by performing the same tests using different cutoffs for minimum number of reads required to register a retained intron. When we only consider full-length reads or increase the stringency for number of mapped reads, the absolute number of transcripts detected falls, but the rate of alternative splicing does not change, demonstrating that our conclusion is not an artefact of very-low-frequency events and low sequence coverage. We have added to lines 284-285 and lines 290-297 as the result of the additional analyses. We have also edited line 340 to make it clear that the FLAIR analyses for productivity only considers full length reads as defined by the tool. Line 331-332 has been edited as suggested.

Minor comments:

1. What would the justification be for the use of the ME49 strain Illumina dataset as compared to sequencing the Pru strain on the Illumina platform, besides a relative lack of comprehensive datasets? This is for a fair comparison as opposed to a ballpark, as the reasoning was that the ME49 strain is closely related to the Pru strain, but the degree of similarity between the Pru and ME49 transcriptomes has not been pointed out.

- We characterized Prugniaud (Pru) by nanopore sequencing as a type II strain that is highly competent to differentiate into bradyzoites, and for its potential for future analyses investigating transcriptional changes in bradyzoites. We used the existing Illumina data from ME49 strain as a comparator because it has the highest coverage and best quality annotated *Toxoplasma* transcriptomics dataset available for comparison. Of the well-characterized strains, TgME49 is the most closely related to Prugniaud (Pru) with very low genetic variation between the two strains (less than 0.01%). We have amended the manuscript to explain this (lines 226-227).

2. The point about rRNA reads present in Illumina versus nanopore datasets- would this be due mainly to the fact that the RNA was directly sequenced and therefore sequences without polyA tails were not sequenced, and not what the authors point out to be differences in polyA purification or PCR amplification bias?

- Yes, this is an important question that we hadn't sufficiently addressed in the first version. The addition of the sequencing adapters during the preparation of material for nanopore sequencing does also likely reduce the sequencing of RNA molecules without polyA tails. The library prep for ONT sequencing as shown in the schematic below means that only polyA tail RNA is ligated to the adaptor for sequencing. We have changed the wording on line 251 to reflect this.

Reviewer #2 (Comments for the Author):

The paper by Lee et al. entitled "Direct nanopore sequencing of mRNA reveals landscape of transcript isoforms in apicomplexan parasites" describes the results obtained by using Oxford NanoPore Seq of two Apicomplexan parasites: *Toxoplasma gondii* and *Plasmodium falciparum*. The main finding of this ms is that intron retention is the most common alternative splicing event in these parasites.

The paper is clearly written (in most parts), nicely structured and the data is well presented. Introducing Nanopore sequencing, which enables analysis of long transcript reads has obvious potential for detection of alternative splicing events as the outcome is entire or almost complete transcripts. It also had the potential to provide a valuable comparison between illumina and nanopore sequencing for analyzing alternative splicing events in *T. gondii* and *P. falciparum*. However, the experiments performed seem preliminary. They present the results of only one run of unsynchronized parasite culture. The quality of the data obtained appears to be low, less than 50% of the plasmodium transcripts aligned properly to the reference genome. The threshold used is 3 transcripts per gene which is very low. As such, it is hard to perform proper comparison to illumina technology (I also could not find the exact reference for the illumina data set used).

From a biological aspect the paper does not add much to what was already known (intron retention appeared also to be the most abundant alternative splicing event in previous publications). Few of the previous publications (some already 10 years old) that used illumina technology and first describe the 977 alternative splicing events in blood stage parasites (more than what was found in the current papers) are not mentioned.

- While less than 50% of the *Plasmodium* transcripts aligned properly to the reference genome, this was not the case for *T. gondii*, strongly suggesting that the AU rich nature of *Plasmodium* hinders alignment, rather than low quality. We inspected the unmapped sequence and found that they were AU rich *Plasmodium* transcripts, but with low confidence scores. This is also a problem for Illumina where many AT rich segments map non uniquely to the *Plasmodium* genome. We have addressed this in the discussion (lines 477-485).

The choice to use 3 as a threshold is based on several reasons. We have tested multiple thresholds and find that for reads numbers per gene (or event) over 3, the patterns observed and therefore conclusions did not change. As part of the response to reviewer one described above, we have now included an additional analysis showing that while absolute numbers of transcripts defined as detected rises when using a lower read requirement, the patterns of splicing observed do not. We have made this clearer in several places. While a threshold of 3 reads for an illumina analysis would be very low, a threshold of 3 in our nanopore analysis represents 3-6 kb of data based on the average read lengths. For Illumina sequencing, that would represent 10-20 reads. The third reason we consider this to be an appropriate threshold is that previous publications, including one specifically looking at intron retention

- <https://genomebiology.biomedcentral.com/articles/10.1186/s13059-017-1184-4> [1], uses a threshold of 3 as well. Reference for the Illumina data has now been added on line 650.

Regarding the reference to previous publications, we assume the referee is referring to the Sorber, Dimon and DeRisi (2011) [2] paper which identified 977 “events”. However, we suspect this is a misinterpretation of that phenomenon as the paper identified 977 new canonical splicing events (absent from gene model), not alternative splicing. In the paper “310 alternative splicing events were detected in 254 (4.5%) genes, most of which truncate open reading frames”. We find it very difficult to directly compare alternative splicing events between publications due to differences in the methodologies and changes in the gene annotation over time but we have now done the comparison looking at gene overlap between the 2 datasets (lines 435-438). We were not clear in our initial transcript that the numbers we presented represented alternatively spliced genes, not alternatively spliced events. We have now edited lines 409-410 to include this number. The novelty of this paper over previous publications is that we are directly detecting alternatively spliced events from long read RNA and not fragmented cDNA. Previous publications have shown that alternatively spliced events are likely to lead to unproductive transcripts but simultaneously occurring alternative splicing events may affect the results, which would not be easily detected in short read sequencing. Our full-length isoform data provides direct evidence for the notion that these events occur in mature, complete mRNA molecules and are indeed non-productive.

Reviewer #3

Lee and colleagues here present a first comparison of Illumina short-read sequencing and Oxford Nanopore Technologies (ONT) direct RNA long-read sequencing with two apicomplexan parasites: *Toxoplasma gondii* (Tg) and *Plasmodium falciparum* (Pf). Using readily produced Tg tachyzoites from the Pru strain, and Pf mixed blood stages from the common lab strain 3D7, the authors put these samples through the manufacturer's recommended workflow and compare these data to published Illumina datasets. Using these comparisons, they focus on differential splicing events that occur in these related apicomplexan parasites, which is a strength of the ONT sequencing platform due to the long reads it provides. As has been seen for other eukaryotes, these long reads often provide full-length (or nearly full length) transcripts and provide a wealth of direct information about splice isoforms. In contrast, Illumina short-read sequencing must rely upon predictive algorithms to anticipate splice isoforms.

One of the major findings presented in this manuscript is the greater abundance of intron retention events as compared to other eukaryotes. What is not clear here is whether all of these mRNAs are those that have completed nuclear processing and have passed quality control to be exported to the cytosol, or if these represent mRNAs that are still resident in the nucleus and are currently undergoing maturation. While poly(A) selection will promote capture of mature mRNAs in most species (as polyadenylation of transcripts typically follows after co-transcriptional capping and splicing), the higher A/U content of *Plasmodium* mRNAs reduces the efficacy of this approach. It would be important to describe how this possibility is excluded (experimentally or otherwise) when other eukaryotes have been studied.

Generally, the sequencing, mapping, and comparison efforts are of very high quality. Most figures are clear and present the data in a concise manner. Comments are provided below that will further clarify the manuscript so that readers more readily follow the data and interpretations, and so that the experiments and analysis can be accurately repeated by others. This first foray into ONT direct RNA sequencing of apicomplexans will be a solid study for mSystems.

Major Points:

-It appears that all of this data is generated from sequencing of a single biological replicate. Most transcriptomic studies require multiple biological replicates for rigor.

> We quite agree that differential expression analysis requires multiple independent biological replicates for robust statistical analysis of transcriptional data, but that is not what we are presenting in this study,

and nor do we represent it as such in the manuscript. The manuscript accurately describes the analysis as a “survey” of the alternative splicing landscape and does not inappropriately attempt analyses that would require replicates for robust statistical analysis.’ Elsewhere we have conducted exactly that type of analysis and have included data from multiple replicates (eg Yeoh et al, Genome Biology 2019, Yeoh et al, BCM Genomics 2017). Some additional experiments that would have required multiple replicates that we had earlier planned had to be cancelled due to the COVID-19 based close down of our laboratories and culture facilities for much of 2020.

- As noted above, intron retention could reflect the sequencing of pre-mRNAs that are still undergoing maturation in the nucleus. Additional discussion about how this could be excluded (if appropriate), the extent that this is observed in transcriptomes of other eukaryotes, or other qualifiers would be helpful. This is especially important due to the claim that this type of splice isoform is more abundant in apicomplexans as compared to other eukaryotes.

> In addition to poly(A) selection, the library prep for ONT sequencing has an additional step for the specific capture of only poly(A) tail RNA (see response to reviewer 1; additional detail added to line 385-386). Previous publications have shown that polyA selection is sufficient to exclude nuclear unprocessed RNA [3]. To exclude the possibility of reduced efficacy of polyA selection for the AU rich RNA of Plasmodium, we looked at gene body coverage of reads. The results, which we describe in line 393-397 indicate that we are indeed sequencing reads captured at the 3' polyA tail end. This is further supported by our deep sequencing of mitochondrial mRNAs (known to carry polyA tails), but the absence of sequence reads for the more abundant apicoplast mRNAs which are AU rich, but which are known to lack polyA tails. This additional information has been added to Figure 2B and lines 254-258.

- Expansion of the bioinformatic processing of the ONT sequencing data is needed so that this work can be fully replicated. For instance, no information is provided as to what Q score threshold is used to designate "pass reads," which has implications for other analyses in this work. It wasn't explicitly mentioned if adapters were trimmed (not doing so would make the data quality lower). Other tools are mentioned, but their use/parameters are not described in detail. An expansion of the explanation of the statistical analyses done using command line/RStudio and where their implementation as represented in the results section is warranted. Inclusion of a description of all methods used in the Results is needed.

> We have added a supplementary file with the commands/parameters and description of each tool used in the analyses (supplementary file SF1, reference added to lines 661-662, 719-720). We have also explained some of the methods in more detail in the file. The command line interface/RStudio was primarily used for data wrangling/post-processing, the results of which were viewed with ggplot (wording changed on line 652-653).

On the point on “pass reads”, pass/fail reads are automatically designated by the software, which is proprietary of ONT (clarified on line 641). They do not specify the criteria or threshold in the documentation. Adapters were not trimmed, and we have no accurate way of doing it to our knowledge. We spoke to the author of Porechop, a software which allows the trimming of adapters in Nanopore DNA reads, who confirmed this. The adapter sequence is DNA, while the sequencing material is RNA. Due to the difference and homopolymer effects, the basecaller is unable to call the adapter sequences accurately. This is also the reason we can be confident that we are not analysing contaminating genomic DNA because DNA signals simply cannot be interpreted under RNA presets.

Minor Points:

- Line 22: Should be AU content, as this is RNA.

> This has now been corrected

-Line 38: The claim that this is a complete analysis of parasite transcriptomics in the "Importance" section is of course a bit too large and broad. This is an important first step, but many other aspects of parasite biology remain to be explored. Please temper this claim.

> Apologies – we agree that this was quite a very poorly framed over-claim that was indented to reflect the complete transcripts, rather than complete analysis. We have reworded the sentence to “We have used a novel nanopore sequencing technology to directly analyse parasite transcriptomes”.

- Additional/different citations are needed in a few critical locations. This includes:

1) providing the citation for the published Illumina datasets (e.g. in the methods section) is helpful to include in addition to providing the SRA accession numbers.

2) Line 51: Citing the most recent World Malaria Report would be better than a review from a few years back.

3) Line 66: only a citation for Plasmodium is provided here, but a statement is made to encompass apicomplexans.

4) Line 125: Providing a citation on detecting modified bases by ONT direct RNA sequencing in the Introduction is warranted as well (e.g. Liu et al. Nature Communications 2019 may be suitable here).

5) Line 128: Please confirm that Ref 33 is not published (this is a biorxiv preprint from 2019).

-Line 171 and/or Line 422-423: It would be useful to hear more speculation as to why the error rate was so high in these datasets. Could it be due to not trimming adapter sequences? Something else?

> These points have now been corrected. For the last point, the high error rate is mainly attributed to the design and basecalling capabilities of ONT sequencing. Reads are sequenced multiple bases at a time, and ONT utilises neural network to untangle the signals for basecalling. This requires internal development time on ONT's side, as has been shown and discussed for ONT DNA sequencing. RNA sequencing was introduced later and has the additional complication of RNA modifications, but we do expect it to catch up in the future.

Figures and Tables:

(NOTE: The numbering of supplementary figures/tables have changed to suit mSystems requirements)

-Figure 1B: Designation of error bars is not provided (confidence intervals?). Also, the red box plots are not provided for reads >8000, but Quality scores are.

> We have now clarified all information in the figure legend. The red box plots are missing due to low number of reads exceeding 8kb, and thus insufficient sampling to draw informative descriptive statistics.

-Figure 2A: These comparisons are not with matched samples, so it is difficult to glean much from these scatter plots.

> While the comparisons are not with matched samples, it's a proxy to understand how the ONT dataset might compare with an Illumina dataset. This is not a quantitative differential expression analysis and we're not advancing ONT for fine scale differential expression analysis. This is instead a semi-quantitative observation that the gene expression profiles are similar, and indicates that read number correlates well between illumina and ONT for most genes.

-Figure 2B: No y-axis (radial) values are provided, so the circos plots are not informative enough. An inset of the mitochondrial RNA (currently not shown) and the apicoplast RNA (not shown in a useful way) would be helpful.

> Thanks for picking up this omission. We have now included y-axis values and an inset of the mitochondrial + apicoplast RNA. As mentioned above, the additional inset reveals high proportion of reads mapping to the mitochondrion but not apicoplast, which is additional evidence we are selectively capturing full length, polyA RNA (known to be present on mitochondrial but not apicoplast mRNA).

-Figure 3B: There is an additional color in the bar chart (light red) that is not designated.

> The light red is the overlap between the red and grey background.

-Figure 3C: Are tables allowed in Figures for mSystems?

> The instructions to authors specify that Tables with complex shading should be included as figure elements and we believe that this figure therefore complies with the instructions.

-Figure 3D: As noted above, the negative correlation is clear for Plasmodium, but not for Toxoplasma. Please solidify or temper this claim.

> We have modified this claim on line 352

-Figure 4: It would be helpful to note the types of events that are being binned as "uncharacterized" here.

> This is noted on lines 412-416. We have now added the detail to the figure legend.

-Supp Figure 1: The reported Quality score range here is discordant with the 20% error rate reported.

> Quality scores are based on the internal basecalling metric while error rate is based on accuracy of the read to the annotated genome file. They are therefore not directly comparable. A more comparable link within the figure is percent identity.

-Supp Figure 2: These comparisons are not very useful, as they are comparing mixed stage parasites with one sequencing method with synchronized parasites with the other. Consider omitting this.

> We feel that this figure provides additional detail about the composition of the mixed stage parasites but will omit it if the editor agrees that it is unnecessary.

-Supp Figure 3: It is likely that this is just a bad annotation for these two genes. Are there other examples of this?

> Yes, this is noted on lines 302-303. We have now listed the exact numbers on line 300. We have also added IGV snapshots of reads that map the gene model accurately as comparison.

-Supp Figure 4: The blue color on the right is not the same as the legend.

> This has now been corrected. We thank the referee for noticing this error.

-Supp Figure 5: There are no confidence intervals provided here, whereas they are provided in a similar figure in the main text.

> The figure in the main text averages multiple genes for each bin. A confidence interval is therefore appropriate and necessary. This supplementary figure calculates the bin at a global scale; the data is aggregated before transformation. This eliminates discrete data points and we can't generate a confidence interval, but it allows us to understand the data in a different way.

-Supp Table 3: The Plasmodium tab is blank.

> We thank the referee for noticing this mistake. It has now been fixed

-Supp Table 6: This is not referenced in the manuscript, and there is no legend for it either. It appears to be just a gene list.

> Apologies – we must have made an error with the manuscript handling system - there should be no Supp Table 6. and we have now removed that file. The gene list is supp. Table 5, which was referenced in the manuscript. This table is now supp. Table 4 to fit with mSystems requirements.

Typos/Grammar:

Line 22: Italicize Plasmodium

Line 29: "particularly intron retention" instead of particular intron retention

Line 90: "is" instead of "in"

These errors have now been corrected.

1. Middleton, R., et al., *IRFinder: assessing the impact of intron retention on mammalian gene expression*. Genome Biology, 2017. **18**(1): p. 51.
2. Sorber, K., M.T. Dimon, and J.L. DeRisi, *RNA-Seq analysis of splicing in Plasmodium falciparum uncovers new splice junctions, alternative splicing and splicing of antisense transcripts*. Nucleic Acids Res, 2011. **39**(9): p. 3820-35.
3. Zhao, S., et al., *Evaluation of two main RNA-seq approaches for gene quantification in clinical RNA sequencing: polyA+ selection versus rRNA depletion*. Scientific Reports, 2018. **8**(1): p. 4781.

February 6, 2021

Dr. Stuart A Ralph
University of Melbourne
Department of Biochemistry and Molecular Biology
Bio21 Molecular Science and Biotechnology Institute
Parkville 3010
Australia

Re: mSystems01081-20R1 (Direct nanopore sequencing of mRNA reveals landscape of transcript isoforms in apicomplexan parasites)

Dear Dr. Stuart A Ralph:

Your manuscript has been accepted, and I am forwarding it to the ASM Journals Department for publication. For your reference, ASM Journals' address is given below. Before it can be scheduled for publication, your manuscript will be checked by the mSystems senior production editor, Ellie Ghatineh, to make sure that all elements meet the technical requirements for publication. She will contact you if anything needs to be revised before copyediting and production can begin. Otherwise, you will be notified when your proofs are ready to be viewed.

- Minimum resolution of 1280 x 720
- .mov or .mp4. video format
- Provide video in the highest quality possible, but do not exceed 1080p
- Provide a still/profile picture that is 640 (w) x 720 (h) max

We recognize that the video files can become quite large, and so to avoid quality loss ASM

suggests sending the video file via <https://www.wetransfer.com/>. When you have a final version of the video and the still ready to share, please send it to Ellie Ghatineh at eghatineh@asmusa.org.

Sincerely,

Paola de Sessions
Editor, mSystems

Journals Department
Table S4: Accept
Fig S4: Accept
Fig S2: Accept
Table S3: Accept
Table S2: Accept
Fig S3: Accept
SF1: Accept
Table S1: Accept
Fig S1: Accept
Fig S5: Accept